# Distance Measurement of a Frequency-Shifted Sub-Terahertz Wave Source

**Minoru Honjo** [1] , **Koji Suizu** [1,*], **Masaki Yamaguchi** [1] **and Tomofumi Ikari** [2]

[1]  Department of Information and Communication Systems Engineering, Chiba Institute of Technology, 2-17-1 Tsudanuma, Narashino 275-0016, Japan; s16a5109vl@s.chibakoudai.jp (M.H.); ma_yamaguchi@nohmi.co.jp (M.Y.)
[2]  Spectra Design Ltd., 285-1 Yuzukami, Otahara 324-0403, Japan; ikari@spectra-dsn.co.jp
*  Correspondence: suizu.koji@p.chibakoudai.jp

**Abstract:** In this paper, we report the development of a frequency-shifted (FS) terahertz (THz) wave source for the non-destructive inspection of buildings. Currently, terahertz-time domain spectroscopy (THz-TDS) is the mainstream method for non-destructive inspection using THz waves. However, THz-TDS is limited by its measurement range and difficulties encountered when there is a strong frequency dependence in the absorption characteristics and refractive index of the measurement target. To address these issues, we developed a novel non-destructive approach for inspection applications using frequency-shifted THz waves. Our system uses a frequency-shifted feedback (FSF) laser as the pump light source to generate FS-THz waves; this allowed us to obtain precise distance measurements of objects over a broad range of distances. We tested a prototype FS-THz system and confirmed successful measurement of spatial distances inside a building material.

**Keywords:** distance measurement; FSF laser; FS terahertz waves; non-destructive inspection; OFDR

## 1. Introduction

The aging of building structures is an ongoing problem. Accidents caused by the deterioration of structures, for example during earthquakes, highlight the fragility of these structures. To prevent these accidents, it is important to evaluate temporal changes quantitatively using rapid physical measurements. The establishment of remote and structural inspection technology would provide solutions to this end and contribute greatly to the construction of a safe and secure infrastructure. Electromagnetic waves can be used effectively for remote non-destructive measurements. Terahertz (THz) waves, which lie between radio waves and far-infrared light on the electromagnetic spectrum, are of particular interest, as they have high transparency to materials, high spatial resolution compared to microwaves and a lower photon energy; as such, they are being used in non-destructive inspection applications [1–3].

Currently, terahertz-time domain spectroscopy (THz-TDS) is the main approach for non-destructive testing using THz waves and various applications have been developed in recent years [4–10]. THz-TDS can be utilized to collect distance information based on the time required to receive the reflected pulse under ultrashort pulse irradiation. To measure long distances during sampling using ultrashort pulses, it is necessary to perform scan steps in the order from microns to tens of centimeters in length. Thus, there is a trade-off between the depth resolution and measurement time. For example, an optical path delay length of several meters requires very long measurement times. Therefore, it is difficult to detect when a defect is located more than a few centimeters from the wall surface. Furthermore, if the absorption characteristics and refractive index of the material have a strong frequency dependence, the pulse waveform changes significantly. In such a situation, the pulse waveform collapses, making position measurements difficult. These problems cannot solve and greatly inhibit application to building inspections.

Non-destructive inspections can also be carried out using monochromatic THz waves. Specifically, continuous THz waves irradiate the measurement target and the reflected and transmitted THz wave intensities are recorded. Given that the received intensity depends on the absorption and scattering characteristics of the material being measured, this method provides the means to visualize the interior of the material. In addition, if the material has cracks, the reflection intensity changes due to scattering, thus allowing for defect analyses of the structure to be carried out. Notably, this method provides only two-dimensional images of the reflection and transmission intensities, without depth or phase information [11–14]. In the frequency-modulated continuous-wave (FMCW) radars in the THz band, there are methods such as multiplying frequency-swept microwaves to generate THz waves [15] and generating THz waves by the difference frequency of two continuous-wave lasers, one having the frequency swept and the other having a fixed frequency laser light source [16]. FMCW radars in the THz range have been used for concealed object imaging and thickness measurement of materials. The frequency sweep speed of such conventional FMCW radars is from several hundred to several THz/s.

In this paper, to develop a real field inspection system for buildings using THz waves, we used a frequency-shifted (FS) THz source to create a precise non-destructive distance measurement device. Currently, a precise distance measurement of optical frequency-domain reflectometry (OFDR) measurements using a frequency-shifted feedback (FSF) laser is being developed [17–20]. By combining the superiority of OFDR measurements with an FSF laser as a distance measurement method with the transmission of THz waves to various substances, our system enables the non-destructive inspection of objects that were difficult to measure in the past (e.g., defect inspections of buildings, structural inspection of dielectric substrates and plastic defect inspections). FS-THz waves have an ultra-fast frequency sweep of several PHz/s. In addition, the chirped comb structure enables detection of higher-order beats in OFDR, compared to the conventional frequency-swept THz wave measurement, resulting in high-speed and high-accuracy distance measurements. We used the created experimental system to measure distances and show that distance measurement using FS-THz waves could be applied to real field inspections of buildings.

The remainder of this paper is organized as follows: Section 2 introduces the principle of OFDR distance measurement. Section 3 describes the generation of FS-THz waves and the distance measurement process using FS-THz waves. Section 4 presents the experimental setup that combines FS-THz wave generation with OFDR measurements inside a building material. Section 5 presents our results and Section 6 concludes with directions for future work.

## 2. Optical Frequency Domain Reflectometry by Frequency-Shifted Feedback Laser

An FSF laser has an acousto-optic modulator (AOM), which is an optical frequency shifter, incorporated in its laser cavity such that the intracavity field frequency is shifted by a fixed amount at every round trip transit. Figure 1 presents a schematic of a chirped comb of an FSF laser. Instantaneous frequencies are present at each cavity longitudinal mode frequency interval. The left side is the light intensity and shows the oscillation spectrum. The right side shows the time variation of the instantaneous frequency with a slope $\gamma$ and $1/\tau_{RT}$ is the cavity free spectral range (FSR). These are present in a comb-like pattern for each free spectral width. $\gamma$ is also the chirp rate (frequency sweep speed), typically several PHz/s, which is very fast [18].

In OFDR measurements using an FSF laser, the frequency difference is obtained as a beat signal, corresponding to the optical path difference. Figure 2 presents a schematic diagram of the principle of OFDR distance measurements using a Michelson interferometer. An FSF laser light source is injected into the Michelson interferometer and split into reference and signal beams. The reflected beams are then merged. If there is an optical path difference, a frequency beat is obtained. The optical path difference can be obtained by analyzing the beat frequency [21,22].

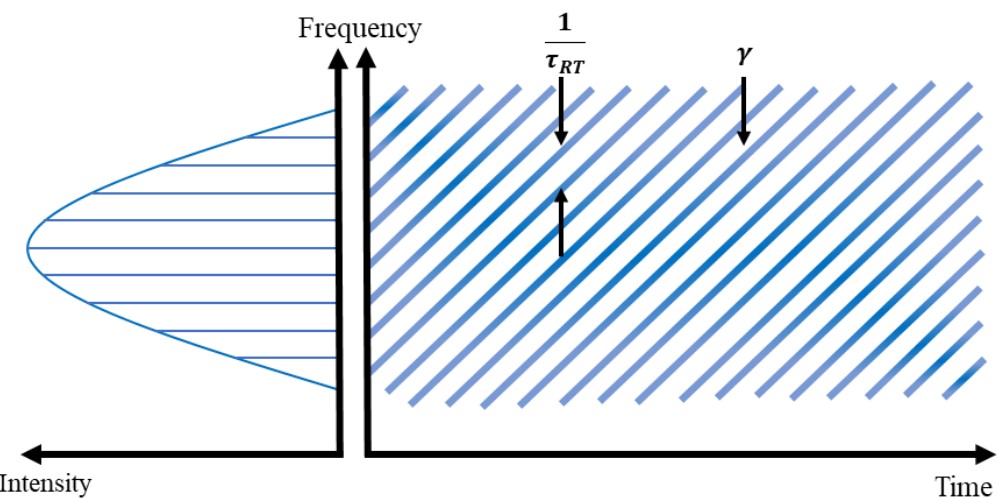

**Figure 1.** Schematic of a comb of chirped frequencies.

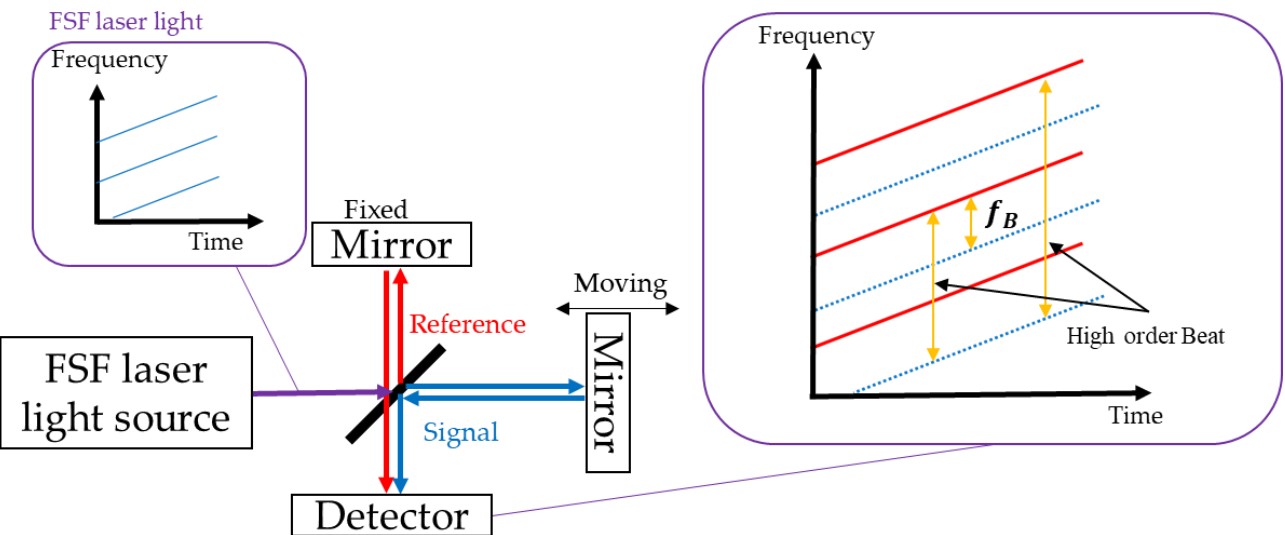

**Figure 2.** Optical frequency-domain reflectometry (OFDR) setup with a Michelson interferometer.

Given a refractive index of the propagated light path of $n$, an optical path difference of $L$ and the speed of light in vacuum $c$, the time difference $\tau$ between the reference beam and the signal beam can be expressed as [23]

$$\tau = \frac{nL}{c}. \tag{1}$$

If the beat component, due to the optical path difference, exceeds the frequency mode of the resonator, the same distance measurement is possible by observing the higher-order beat signal. In principle, the measurement accuracy in this case can be extracted from the chirped frequency of the FSF laser, as the reference frequency source. Using this principle, it is possible to observe large, complex-shaped objects. In addition, the output spectrum of the FSF laser is a comb of chirped frequency components; therefore, multiple beat signals are generated simultaneously. The beat frequency $f_B$ of OFDR measurements using an FSF laser can be expressed as

$$f_B = \gamma\tau - \frac{q}{\tau_{RT}}, \tag{2}$$

where the chirp rate $\gamma$ of an FSF laser is determined by AOM frequency and the cavity round-trip time $\tau_{RT}$, $q$ is the beat order. In the case of the FSF laser used in this study, a

cavity FSR of 40 MHz, a 7.5 m cavity and an AOM frequency of 55 MHz provided a chirp rate of 4.4 PHz/s [24]. The wavelength sweep range was 0.68 nm. The beat frequency in terms of the optical path difference $L_0$ is given by

$$f_0 = \gamma \frac{L_0}{c}, \tag{3}$$

with an optical path difference of $\Delta L$, the expression for the beat frequency is as follows:

$$f_1 = \gamma \frac{L_0 + \Delta L}{c}. \tag{4}$$

the difference in beat frequencies can be expressed as

$$\Delta f = f_1 - f_0 = \gamma \frac{\Delta L}{c}. \tag{5}$$

Therefore, the relationship between the frequency variation and optical path difference in OFDR using an FSF laser is given by

$$\frac{\Delta f}{\Delta L} = \frac{\gamma}{c}, \tag{6}$$

for this study, the ratio described by Equation (7) was 14.7 kHz/mm.

## 3. Frequency-Shifted Terahertz Waves

The FSF laser light and monochromatic laser light are combined as the pump light, such that the difference in their center frequencies creates the THz waves. The THz waves have a THz period beat corresponding to the difference between the center frequencies of the two lasers, thus creating the chirp frequency. When the pump light is injected into a uni-traveling-carrier photo diode (UTC-PD) [25], a beat-driven current is generated and FS-THz waves having the characteristic chirp frequency are emitted. Figure 3 presents the spectrum of the FSF laser light and the monochromatic laser light used to generate FS-THz waves. The difference frequency (DF) between the FSF laser light and the monochromatic laser light was set to 210 GHz, which is the center frequency of FS-THz waves. The THz frequency of the FS-THz waves can be selected by sweeping the wavelength of the monochromatic laser light. Figure 4 presents the output voltage obtained using a Golay Detector (GOLAY CELL GC-1P) plotted for every 0.1 nm in the wavelength-tunable range of FS-THz waves from 0.1 to 1.0 THz. The appropriate frequency for the measurement target can be selected from this frequency range.

The distance resolution of OFDR using FS-THz waves depends on the FSF laser, as follows:

$$\Delta z = \frac{c}{2B} \tag{7}$$

where $B$ is the half-width of the spectrum of the FSF laser [24]. In this study, the half-width of the FSF laser was approximately 84 GHz, as shown in Figure 3, which means that the distance resolution was approximately 1.8 mm.

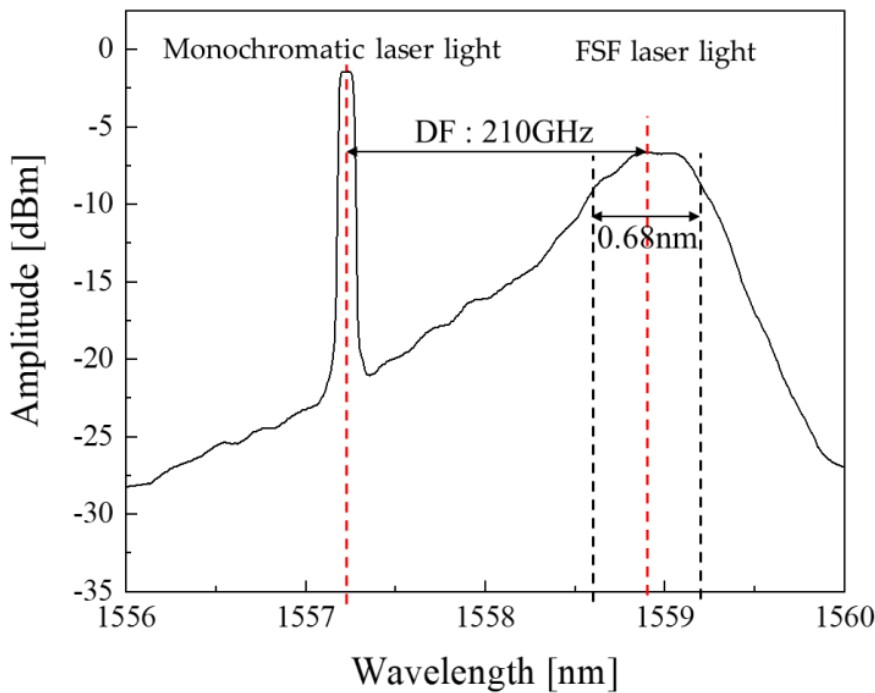

**Figure 3.** Pump light spectrum for generating frequency-shifted terahertz (FS-THz) waves.

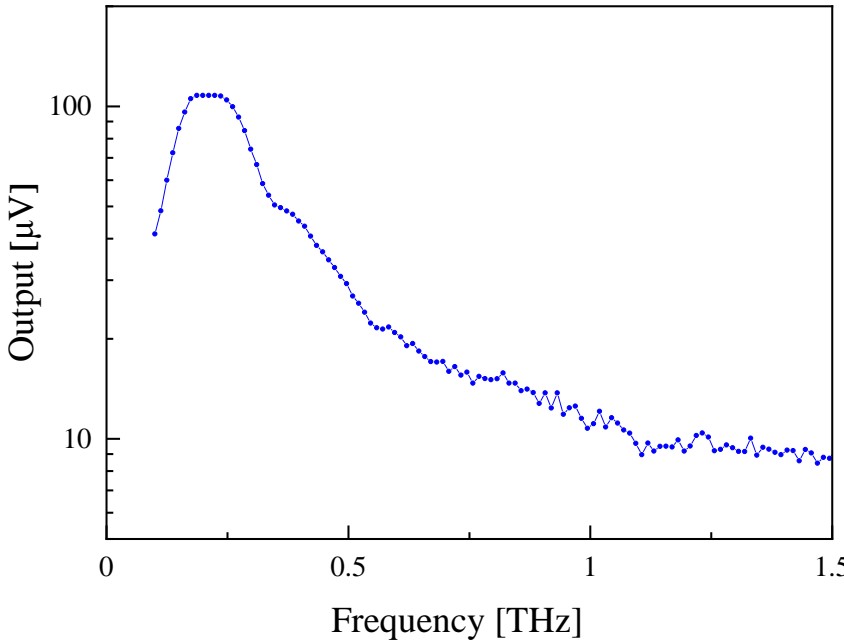

**Figure 4.** Frequency characteristics of the FS-THz intensity.

## 4. Distance Measurement Experiments Using Frequency-Shifted Terahertz Waves

We conducted experiments using the prototype system presented in Figure 5. The FSF laser light and the external cavity laser diode (ECLD) were amplified by an erbium-doped fiber amplifier (EDFA) and combined. The combined dual wavelength light was directed into a UTC-PD, resulting in difference frequency generation via photomixing to create the FS-THz waves. The frequency of the FS-THz waves was set to approximately 170 GHz and was amplified further by a 170–260 GHz amplifier (VDI4.3AMP-0170 THz amplifier). The THz waves' power without the THz amplifier was approximately 76 μW and, with the THz amplifier, was approximately 19.5 mW, we confirmed an amplification of approximately

24 dB. The FS-THz waves were then injected into the Michelson interferometer and separated into signal and reference light. After traversing their optical paths to create an optical path difference, the light was recombined. The combined FS-THz waves have a frequency difference that depends on the optical path difference in the Michelson interferometer. One mirror in the optical path of the interferometer is fixed in position and the other mirror can be moved. By changing the position of the movable mirror, the optical path length can be varied. Once detected by the Schottky barrier diode (SBD) detector, a current is generated with an RF frequency component corresponding to the pumping optical waves. This RF frequency component has the characteristics of the FSF laser light (as the pump light source).

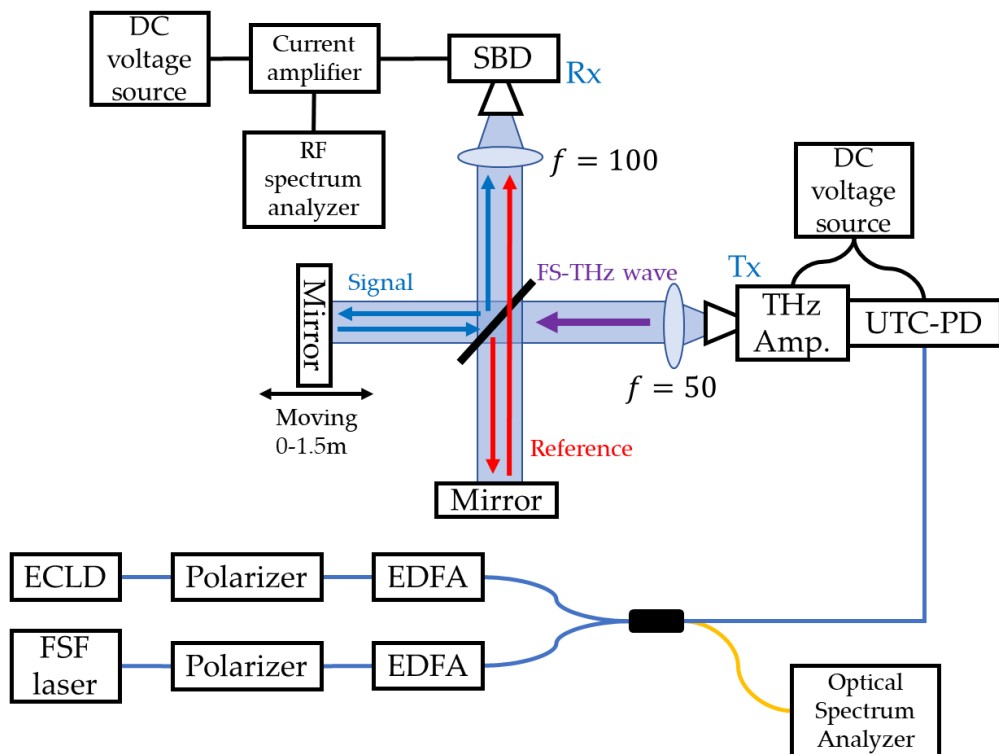

**Figure 5.** Experimental system of distance measurements using FS-THz waves.

The beat signal appears in the range of 0–40 MHz depending on the optical path difference. Second-order and higher-order components are present in the frequency bands of 40–80 MHz, 80–120 MHz and beyond. We observed the spectrum of this output current using an RF spectrum analyzer and confirmed the generation of the beat signal. The optical path difference was then determined based on the frequency of the signal.

We inspected the internal structure of a ceramic tile similar to those used in homes and bathrooms (Figure 6). The mirror on the signal light side was replaced with the tile for the measurement. Figure 7 presents the behavior of the FS-THz waves reflected from the front and back surfaces of the tile. Two beat signals are generated by the reflection of the THz waves from the two surfaces with respect to the reference THz waves. The thickness of the tile was measured by comparing the two beat frequencies.

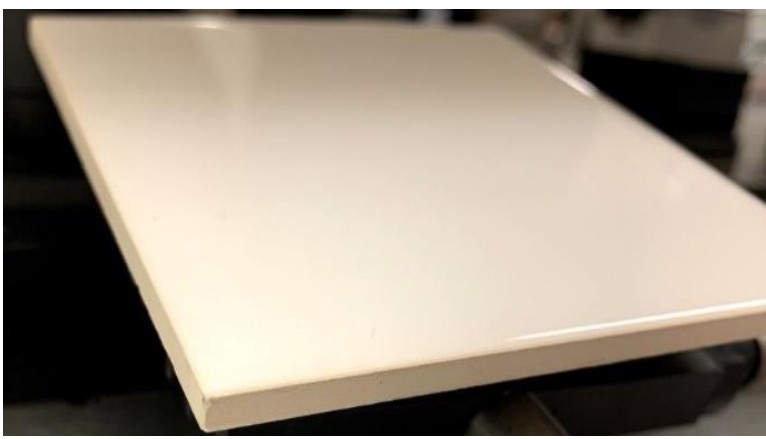

**Figure 6.** Photograph of the tile used in experiments.

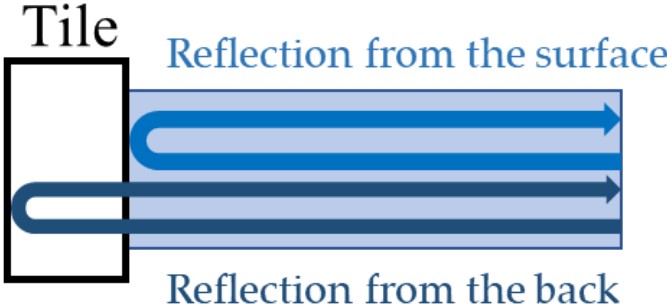

**Figure 7.** Behavior of FS-THz waves in tile thickness measurements.

## 5. Results

Figure 8 presents the results when we generated a beat signal with OFDR using FS-THz waves. The movable mirror was moved every 75 mm. Higher-order beat signals appeared in the 2 GHz band range. In the OFDR measurement using FS-THz waves, a beat signal is generated between the resonance frequencies, depending on the optical path difference. The FSF laser used as the pump light source had a resonant frequency of 40 MHz, i.e., there was a resonant frequency every 40 MHz. The spectrum at both ends of Figure 8 coincides with the resonance frequencies and the spectrum of the beat signal every 75 mm optical path difference appears between these resonance frequencies. At the frequency corresponding to the strongest beat signal, the signal-to-noise ratio was ~17 dB, which was the limit of the beat signal strength of the FSF laser itself. In addition, the THz waves collimating beam had a little spread. This beam spreading appeared as a loss; therefore, the signal intensity decreased by increasing the optical path length difference. We also moved the movable mirror every 75 mm to check the frequency variation of the beat signal. The difference between individual beat frequencies was approximately 2.2 MHz, as shown in Figure 8. The theoretical value was 1.1 MHz when the optical path difference of 75 mm is converted to a frequency, but it was 2.2 MHz because the optical path was round-trip. In terms of the optical path length, we were able to continuously measure a change in the optical path length of approximately 1.36 m. At this time, the light source and the detection unit were not adjusted at all. Thus, detection was possible without adjustment, no matter where the reflection point was in the distance measurement range of approximately 1.5 m. The measurement accuracy of FS-THz waves depends on the measurement accuracy of the FSF laser and is be expressed as

$$\sigma_{nL}^2 = \frac{c^2 \tau_{RT}^2}{f_s^2} \left( \sigma_{f_B}^2 + f_B{}^2 \tau_{RT}{}^2 \, \sigma_{\frac{1}{\tau_{RT}}}^2 \right) + \frac{nL^2}{f_s^2} \, \sigma_{f_s}^2 \qquad (8)$$

where $\sigma_x$ is the error for index $x$ and $f_s$ is the amount of frequency shift of 110 MHz. The measurement accuracy of the FSF laser we used was $+-50$ µm. The measurement distance obtained from the difference beat frequency between the standard point and the +75 mm point was 73.94 mm and, considering the measurement accuracy, it was 73.89–73.99 mm. Since the optical path difference was set at 75 mm, there was an error of 1.01–1.11 mm, but this was considered to be an error caused by the manual movement of the mirror; therefore, the measurement was generally considered to be accurate.

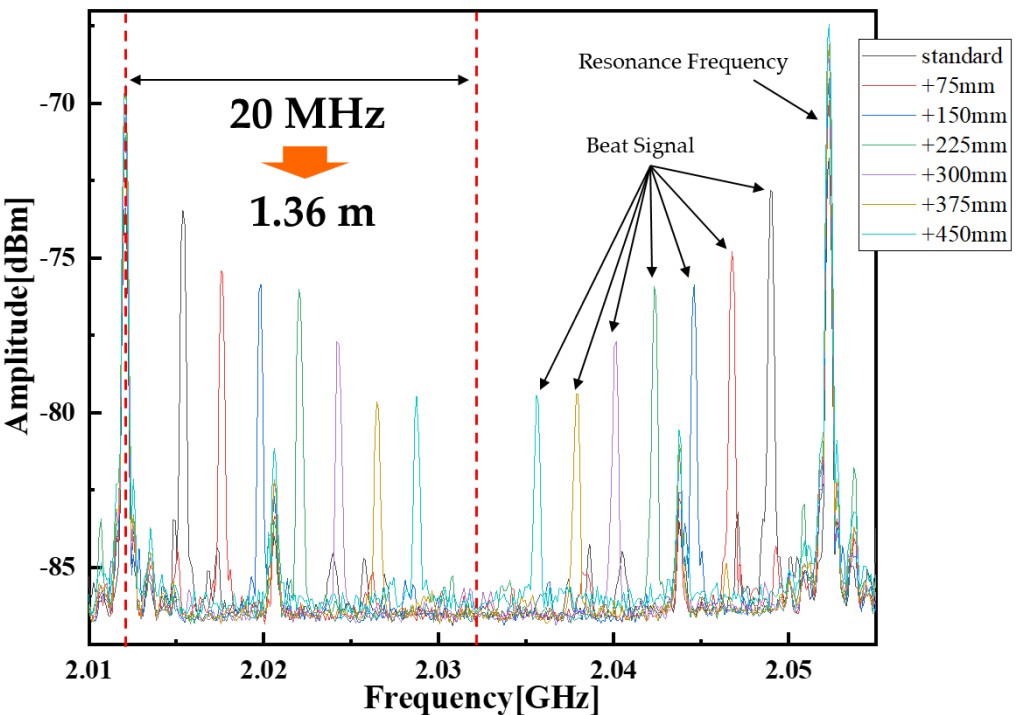

**Figure 8.** Beat signal spectrum and its frequency variation.

Figure 9 presents the beat signal when the tile was measured. Two beat signals can be seen in a single measurement, corresponding to reflections from the front surface and back of the tile. In contrast to the reflectivity of the mirror (100%), the reflectivity of the tile was very small and the loss of that amount appeared. Therefore, the noise level became smaller. The beat frequency by reflection from the front surface was 2.01161 GHz and that from the back surface was 2.01128 GHz. Therefore, the beat frequency difference between the surface and backside reflections was 330 kHz. The obtained beat frequency difference could be converted into a round-trip distance of 22.4 mm, or a distance measurement of 11.2 mm. Considering the refractive index of the tile of 1.88, which was measured by THz-TDS transmission measurements, this corresponded to a distance of 5.96 mm in the interior of the tile. The value measured with a micrometer was 5.11 mm, with an error of 0.85 mm. As a result, the measurement was successful with an error of less than 1 mm. Thus, the distance inside of a substance could be measured by OFDR measurements using FS-THz waves.

The time taken to acquire the data was 19.95 s for Figure 8 and 40.72 s for Figure 9. The measurement time was not fast enough. However, the reason of the slow measurement was to show spectra which had the information of the distance and we used an RF spectrum analyzer to obtain the spectra. We performed similar experiments using a frequency counter as a measuring instrument and were able to acquire data in 0.01 s in this case.

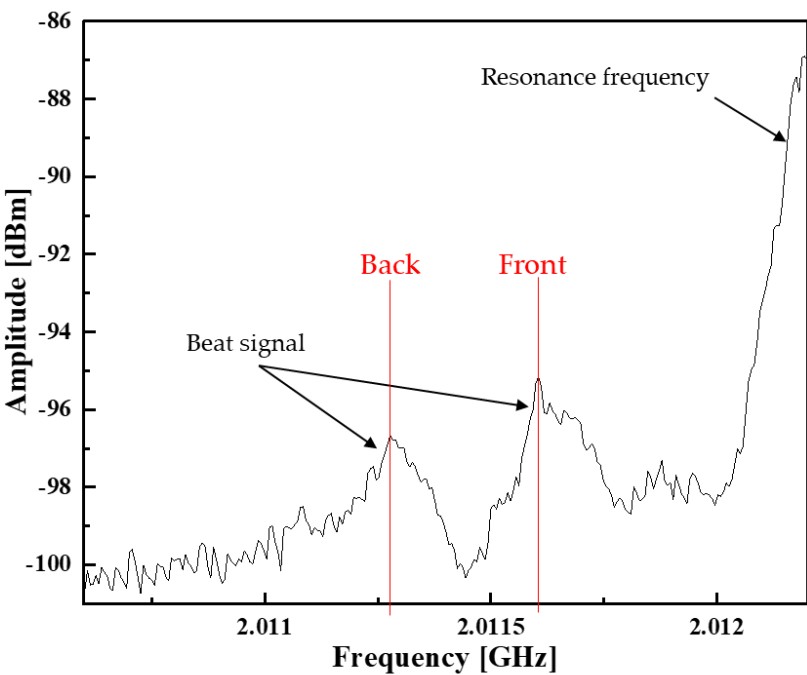

**Figure 9.** Beat signal spectrum generated by reflections from the front and back surfaces of the tile.

## 6. Conclusions and Discussion

We developed a method for the non-destructive inspection of structural building materials using FS-THz wave distance measurements, in which FS-THz waves were generated and measured using OFDR. The FS-THz waves generated had the same characteristics as the pump source, the FSF laser. We performed OFDR measurements of FS THz waves and successfully generated a beat signal with a signal-to-noise ratio of approximately 17 dB, which was the limit of the beat signal strength of the FSF laser itself as the pump light source. We utilized our prototype system to collect long-range measurements of approximately 1.5 m and confirmed its efficacy by measuring the thickness of a ceramic tile with an error of less than 1 mm. Therefore, this method can be used to measure distances inside a material. There is a trade-off between the transparency to materials and resolution, depending on the wavelength. The wavelength selectivity within the proposed approach allows the user to choose the appropriate wavelength depending on the material of the measurement target. Internal measurements can be collected with a wavelength in the 100 GHz band without being affected by the unevenness of the measurement target surface, but defects of a smaller size cannot be detected. Our method also removes the range issues of THz-TDS, allowing the measurement of distances of several meters to be obtained in a single measurement. It overcomes the problem of strong absorption and refractive index effects, as the light source is nearly monochromatic as opposed to a pulse wave. Unlike the method using monochromatic THz waves, our system can also collect depth information, making it possible to measure in three dimensions. When acquiring imaging, the collimated beam diameter corresponds to the special resolution. The collimated beam diameter in this experiment was 10 mm. This can be improved by focusing the beam by placing a lens just before the THz waves are irradiated to the measurement target. Together, our results demonstrate that FS THz waves can be applied to non-destructive internal defect diagnoses of building materials. As a future prospect, we are working on modifying the measurement instrument to be a frequency counter, using the beat frequency of multiple measurement points to visualize the delamination between mortar and tile.

**Author Contributions:** Conceptualization, K.S. and T.I.; Data curation, M.H., M.Y., K.S. and T.I.; Formal analysis, M.H., M.Y. and K.S.; Funding acquisition, K.S.; Investigation, M.H., M.Y. and K.S.; Project administration, K.S.; Supervision, K.S.; Validation, K.S.; Visualization, M.H. and M.Y.;

Writing–original draft, M.H.; Writing–review & editing, K.S. All authors have read and agreed to the published version of the manuscript.

**Funding:** This research was funded by JSPS Grants-in-Aid for Scientific Research (A) grant number JP18H03827.

**Institutional Review Board Statement:** Not applicable.

**Informed Consent Statement:** Not applicable.

**Data Availability Statement:** Raw data that support the findings of this study are available from the corresponding author, upon reasonable request.

**Conflicts of Interest:** The authors declare no conflict of interest.

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
