# Peer review of "Distance Measurement of a Frequency-Shifted Sub-Terahertz Wave Source"

_photonics, doi:10.3390/photonics9030128_

Round 1

Reviewer 1 Report

This paper reports the measurement of tile thickness by frequency-shifted terahertz waves generated by an FSF laser. Non-destructive inspection using the transparency of terahertz waves is desired to be realized due to its spatial resolution. In the OFDR radar method, good resolution can be obtained by using a wide bandwidth. The differential frequency generation using an FSF laser can realize a very wide frequency sweep range and can be expected as a signal source for radar system. In the experiment, the beat signals associated with the reflected waves from the front and back of the tile were measured and identified. However, because the explanation of the characteristics of FS terahertz wave is insufficient, readers cannot fully understand the measurement principle. Also, the measurement results and discussions are inadequate. Therefore, it seems that a major revision is totally necessary to reach the acceptance level. See the detailed comments below.

-Because this paper is about a distance measurement using FS terahertz wave, the title seems inappropriate and should be changed.

-At the introduction, the author mentioned distance measurements using THz TDS system. Please also mention about FMCW radars in terahertz range in the introduction part.

-In Chapter 2, the author mentions the principle of OFDR distance measurement. However, the characteristics of FS terahertz waves generated by the FSF laser are not described in detail. If my understanding is correct, the FS terahertz wave in this experiment should have a multi-peak spectrum instantaneously that corresponds to the longitudinal mode interval of the FSF laser. However, it seems difficult for the reader to imagine the spectrum characteristics because there is no enough explanation. It is also difficult to understand equation (3). Please explain the characteristics of FS terahertz wave and the mechanism of obtaining beat signal using FS terahertz wave sufficiently. I think graphical explanation would be better.

-Related to the above comment, the FSF laser has a multi-peak spectrum instantaneously and the peaks are moving very fast. Fig. 2 seems to be the time average of the intensities of the FSF laser whose spectrum changes over time. Please add such an explanation. Simply showing this figure will make misleading for readers.

-Please describe the wavelength sweeping range of the fundamental mode of the FSF laser used.

-Please explain the definitions of “DF” and “Wavelength” shown in Fig. 2.

-As mentioned earlier, FS terahertz wave has a multi-peak spectrum. Fig. 3 seems to be a plot of the output power of only the fundamental mode, excluding the higher-order mode. Is my understanding correct? If so, please add an explanation on how to measure it.

-For Fig. 7, please explain why two signal peaks appear during the 40MHz span.

-The measurement of Fig. 7 showed that the beat signal changed by the position of movable mirror. Please mention and discuss the measurement accuracy in this measurement.

-In the measurement of Fig. 7, the maximum SN ratio was 17 dB, which was limited by the characteristics of the FSF laser. Is it difficult to improve the signal-to-noise ratio anymore? If so, the applicable target of inspection would be very limited.

-For Fig. 8, does the obtained spectrum change over time? In this measurement, the SN ratio is not good and it seems difficult to read the beat signal frequency accurately.

Reviewer 2 Report

The results presented in the work are quite interesting. In my opinion, there are two points that should be corrected before publishing it.

  1. Basically, the work deals with the radiation of the subterahertz range or the so-called millimeter waves. From my point of view, the proposed approach and terahertz radiation with a frequency > 1 THz are generally quite difficult to use for remote inspection of buildings for several reasons. 1. Strong growth of attenuation by atmospheric water. 2. A strong drop in the efficiency of radiation generation by the proposed method with increasing frequency (as can be seen from Figure 3). 3. Strong absorption and scattering of THz waves by building materials. In this regard, I would suggest changing the title and introduction of the manuscript accordingly, with an emphasis on millimeter waves.
  2. The approach based on the use of single-frequency and swept lasers for generating swept THz radiation is not innovative, as well as its application for thickness measurements. Reports of such systems have been presented at conferences for about 5 years. It is worth noting a fresh paper in which a comparison of the spatial resolution of two systems based on terahertz time-domain spectroscopy and frequency-domain spectroscopy was carried out [https://doi.org/10.1007/s10762-021-00831-5]. Taking this into account, I would like to see in the introduction of the manuscript a greater emphasis on the novelty and advantages of the proposed by the authors quchnique.

As a minor note. It is recommended not to redefine previously entered values. So, for example, on line 78, the parameter gamma (chirp rate) is entered, which is redefined on line 86. Similarly for the light speed constant: lines 75 and 91.

Reviewer 3 Report

Manuscript ID: photonics-1545804

Title: Distance Measurement of a Frequency-shifted Terahertz Wave Source

Honjo et al. report the laser-based generation of frequency-shifted sub-THz wave and proof-of-concept demonstration of non-destructive thickness measurement of a building material, 5-mm-thick ceramic tile. Although the distance measurement principle and experimental results by the developed frequency-shifted THz-wave source are well-written, the presentation quality of the present manuscript is less than satisfactory as a scientific paper. Therefore, for the benefit of the readers, I encourage the authors to make some major revisions to my comments listed below.

1) First of all, because I was confused by some sentences, please ask someone familiar with the English language or use an English editing service to help you revise the manuscript.

2) The authors mentioned in the introduction section that high spatial resolution is one of the big advantages of the use of THz wave. However, for the THz wavelength, there is no focusing element that satisfies such a long standoff distance up to 1.5 m. In the experiment as shown in Fig. 5, the authors used the collimated beam, instead of the focused beam. Therefore, I wonder how will the authors obtain the 3D image of the target that is placed far away from the source with high spatial resolution? Please make sentences on this in the manuscript.

3) Also, from the introduction part, I could not follow the main objective of this study. Please revise the introduction part more logically and clearly.

4) The authors mentioned several times that the frequency sweep rate of the FSF laser is on the order of PHz/s, which is advantageous to reduce the measurement time even in the long standoff distance. However, there is no information on the measurement time that the authors performed. How long does it take to obtain the RF spectra shown in Figures 8 and 9? How fast? Please make sentences on this in the manuscript.

5) I wonder how much THz-wave power was obtained from the UTC-PD emitter with and without a waveguided THz amplifier. And, what kind of detector did the authors use to record the THz-wave intensity shown in Fig. 4. Please comment on these points in the main text.

6) Figure legends for Fig. 8 are somewhat misleading. It should be “standard, +75 mm, +150 mm, … And, why does the signal intensity decrease by increasing the optical path length difference?

7) When one of the mirrors in the interferometer was replaced by a ceramic tile, why the noise level of the spectrum shown in Fig. 9 is much smaller than that of Fig. 8? Is there any difference in experimental conditions?

8) Please add a brief discussion for the improvement of the signal-to-noise ratio of the developed system. I feel that the signal-to-noise ratio of 17 dB is relatively low.

Round 2

Reviewer 1 Report

This paper reports a distance measurement using FS terahertz wave. At the first peer review, I thought that the paper probably included an enough novelty because a wide frequency tunability, which is difficult to be achieved by another THz sources, is possible by the FS terahertz wave and a good resolution/accuracy is expected. To put it the other way, showing good resolution/accuracy with experimental data is the point of acceptance. However, even looking at the response letter, the discussion about the measurement accuracy was insufficient, and it was judged that the measurement was not reliable from the added description. In addition, questions of 8, 10, and 11 were not answered in good faith. I could not find any revisions for these questions in the manuscript. The critical part is as follows.

In the added description related to measurement accuracy, because the author concluded that the measurement error was caused by the manual movement of the mirror, the measurement result shown in Fig. 8 cannot be used for the discussion on the measurement accuracy and discussion using eq. 8 was not meaningful. Although the author showed another experiment in Fig. 9, it also cannot be used for accuracy discussion. In such a noisy state, the peak spectrum is considered to fluctuate over time. If this spectrum is obtained instantaneously, it is very dangerous to discuss and conclude the error with only this spectrum used. If it is a time-integrated measurement, the measurement conditions should be shown in detail, but, there was no description.

As mentioned above, I understand that the advantage of distance measurement with FS terahertz waves is the good resolution/accuracy due to the wide frequency tunability, but this cannot be supported by this experiment and discussion. Therefore, I think this paper should be rejected.

Reviewer 3 Report

The authors have revised the manuscript based on my comments, and therefore, I recommend the publication of this manuscript in its present and revised version in this journal. 

Author Response

Thank you for conducting the peer review.
Also we appreciate your recommendation.

Round 3

Reviewer 1 Report

I have an additional small comment for supplemental data of showing the measurement accuracy. The frequency of beat signal changed with path difference. How did you determine the peak frequency? From the supplemental figure, the linewdth of beat signal seems wide (around several tens kHz). I can also see small peaks probably due to noisy condition. I think the significant figure in the experiment is two.
